



# Not all biodiversity richspots are climate refugia

[1]Ádám T. Kocsis, [2]Qianshuo Zhao, [3]Mark J. Costello and [1]Wolfgang Kiessling

[1]GeoZentrum Nordbayern, Friedrich-Alexander University Erlangen-Nürnberg (FAU), Loewenichstr. 28, Erlangen, Germany
[2]Institute of Marine Science, University of Auckland, Auckland 1142, New Zealand
[3]Faculty of Biosciences and Aquaculture, Nord University, Bodø, 8049 Norway

*Correspondence to*: Ádám T. Kocsis (adam.kocsis@fau.de)

**Abstract**. Anthropogenic climate change is increasingly threatening biodiversity on a global scale. Richspots of biodiversity, regions with exceptionally high endemism and/or number of species, are a top priority for nature conservation. Terrestrial studies have hypothesised that richspots occur in places where long-term climate change was dampened relative to other

regions. Here we tested whether biodiversity richspots are likely to provide refugia for organisms during anthropogenic climate change. We assess the spatial distribution of both historic (absolute temperature change and climate change velocities) and projected climate change in terrestrial, freshwater, and marine richspots. The results suggest that although terrestrial and freshwater richspots have been and will be somewhat less affected than other areas, they are not excluded from the impacts of global warming. Their characteristic biota is expected to witness similar forcing as other areas, including range shifts and

elevated risk of extinction. Marine richspots have warmed even more, have higher climate velocities and are projected to experience higher future warming than non-richspot areas. Our findings emphasise the urgency of protecting a comprehensive and representative network of biodiversity-rich areas that accommodate species range shifts under climate change.





## 1 Introduction

It has been suggested that the reason that some geographic areas are exceptionally rich in biodiversity (sometimes called "biodiversity hotspots") is because they have had little climate change over geological timescales. This long-term stability has led to high numbers of species that are unique to these "climate refugia" (i.e., endemic species; Dynesius and Jansson, 2000; Jansson, 2003; Harrison and Noss, 2017; Senior et al., 2018; Brown et al., 2020). Here, following Manes et al. (2021), we call these areas biodiversity "richspots" to emphasize the distinction from areas of increased temperature, invasive species,

pollution and habitat destruction. If this climate refugium hypothesis is true, then richspots may continue to provide safe harbours (refugia) for species under anthropogenic climate change. Thus, conserving these areas would not only protect species against current human impacts (Halpern et al., 2015; Díaz et al., 2019; Tedesco et al., 2013), such as hunting, fishing and habitat loss, but also limit the effects of climate change on global biodiversity (García molinos et al., 2016).

### 1.1 Biodiversity and climate change

The effects of climate change are now detectable on biodiversity trends since the 1950s (e.g., Chaudhary et al., 2021), and are projected to accelerate in coming decades (e.g. Manes et al., 2021). However, existing human impacts are already devasting biodiversity in all environments. While most confirmed extinctions and threatened species are terrestrial, a higher proportion of freshwater species are threatened, which is reflected in the higher proportion of freshwater richspots impacted by human impacts (Collen et al., 2014; Costello, 2015; Harrison et al., 2018). The rate of species endemism is exceptionally high in

freshwater biogeographic realms, at 89–96% for fish in all but one realm, compared to 11–98% for terrestrial vertebrate groups and 17–84% for marine realms (Leroy et al., 2019). Based on species ranges and conservation status, >25% of IUCN-assessed marine species are threatened in 83% of the oceans (O'hara et al., 2019). The lower thermal safety margins of marine ectotherm species renders them more vulnerable to climate change than terrestrial ectotherms (Pinsky et al., 2019).

### 1.2 Biodiversity richspots

Biodiversity richspots have been proposed in many studies based on different criteria, taxa and geographic contexts (e.g. Myers et al., 2000; Mittermeier et al., 2004; Mittermeier et al., 2011; Asaad et al., 2017; Noss et al., 2015). Eighteen different classifications of marine biodiversity richspots alone have been proposed (Jefferson and Costello, 2019). The most comprehensive scheme of richspots is the so-called "WWF Global 200" (Olson and Dinerstein, 2002; G200), which covers terrestrial, freshwater and marine environments, and has been used in previous climate risk assessments (e.g. Warren et al.,

2018; Manes et al., 2021). In all cases, the delimitation of richspots was based on expert opinion and limited to a few well-known taxa, such as flowering plants and vertebrates. Thus, it is possible that these selections of biodiversity richspots may have taxonomic and/or expert knowledge biases. An objective approach to mapping biodiversity richspots has been applied for the world oceans, using globally standardised data-driven measures of species richness, endemism, habitat, biome and ecosystem distributions (Zhao et al., 2020). This objective designation of representative biodiversity areas (RBAs) indicated



that the 30% most biodiversity rich areas of the ocean would contain 68% of all species, 94% of coral reefs and mangrove forests, and 86% of kelp forests and seagrass meadows.

## 1.3 Climate velocity and range shifts

Climate velocity (Loarie et al., 2009; Burrows et al., 2011) is a key concept to understand the origin and fate of biodiversity richspots under climate change. The velocity of climate change is the pace and direction at which a specified climate variable
changes across geographic space due to changing climate. For example, climate velocity for temperature is the speed at which points of the same temperature (isotherms) move due to changing climate (distance time$^{-1}$). Regions of high climate velocities are those with low topographic relief on land, particularly flooded grasslands and deserts (Loarie et al., 2009), tropical and Arctic regions; as well as offshore tropical and polar regions in the oceans (Burrows et al., 2011; Burrows et al., 2014; García Molinos et al., 2016; Brito-Morales et al., 2018; Brito-Morales et al., 2020).

Some terrestrial areas that have experienced low climate velocities since the last glacial maximum are rich in endemic species and hence more likely to be identified as richspots (Sandel et al., 2011). Climate velocities are also able to predict the direction and pace of past and future species range shifts (Pinsky et al., 2013; Brito-Morales et al., 2018). Marine species tend to follow the physical pathway dictated by climate velocities more closely than terrestrial species probably due to fewer dispersal barriers than on land and the smaller thermal safety margins of marine species (Sunday et al., 2012; Pinsky et al., 2019). Spatial patterns
of climate velocities show regions where species are expected to leave, pass through or arrive within a certain period under a particular climate change scenario (Burrows et al., 2014). Elevated climate velocities are expected to be especially problematic for endemic species, which may have limited dispersal ability (Sandel et al., 2011; Brito-Morales et al., 2018), especially when they live in enclosed seas such as the Mediterranean, from where they can be trapped under global warming.

## 1.4 Anthropogenic climate change in richspots

Current policies put the world on track for around 3°C of heating by the end of the of the century (Hausfather and Peters, 2020). Manes et al. (2021) suggested (based on studies available for half of the richspots), that at this degree of warming, 92% of land-based endemic species and 95% of marine endemics face negative consequences, such as a reduction in abundance. With the doubling of global warming from 1.5°C to 3°C, there is at least a 10-times increase in local extinction risk in biodiversity richspots: rising from 2% for all species on land and sea to 20% and 32% at risk. Of endemic species, 34% and
46% in terrestrial and marine ecosystems, and 100% and 84% of island and mountain species were projected to face high extinction risk, respectively. The fact that these species are endemic suggests they cannot disperse to other areas, and thus a local extinction within a richspot would mean global extinction. In contrast, introduced invasive species were projected to be unaffected by climate change or benefit from it, while their expansion will further threaten the survival of native species. However, if warming rates are lower inside than outside these richspots, then impacts of climate change should be reduced
relative to other regions. Until now, there has been no comparison of recent or projected global warming inside and outside biodiversity richspots. Here we assessed the past and future-projected magnitude of climate change in biodiversity richspots

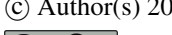



and compared those variables with other regions. We tabulated historic differences in temperatures of the past 50 years and climate change velocities, as well as projected temperature and precipitation change. Our findings are of importance for policies to address biodiversity loss and climate change.

## 2 Data and Methods

### 2.1 Environmental data

### 2.1.1 Observed climatic change and climate velocities

To assess past changes of temperature, we used monthly interpolated data from the CRU TS 4.05 (Harris et al., 2020) and the HadISST1.1 (Rayner et al., 2003) compilations for the near-surface air and ocean surface temperatures (hereafter air and ocean temperatures). We used measurements compiled from the last 50 years to assess the magnitude of change from averages of the 1971–1980 until the 2011–2020 interval. The same timespan (1971–2020) was used to calculate the velocities of climate change for temperature. Climate change velocities were calculated separately for air and ocean temperatures using the "VoCC" R package (Molinos et al., 2019). Original data layers of historic air temperature had $0.5 \times 0.5$; ocean temperature had $1 \times 1$, latitude-longitude degrees of resolution. Antarctica was not represented in the air temperature data.

### 2.1.2 Future climate projections

Future projections of changes of air and ocean temperatures, as well as aggregated precipitation were downloaded from the IPCC Atlas of the AR6 report (Iturbide et al., 2020). These are the results of 6th Phase of the Coupled Model Intercomparison Project (CMIP6, CMIP6, Eyring et al., 2016), and represent multi-model averages at different stages of warming when global warming reaches the +1.5, +2, and +3°C thresholds compared to the simulated pre-industrial baseline (1850–1900). Data layers that represented the same stage of global warming were averaged across four different scenarios (SSP126, SSP245, SSP370 and SSP585) to obtain a single expectation for the three stages of warming. The data also show comparisons to the pre-industrial baseline. Changes of precipitation were rescaled to mm year$^{-1}$. The downloaded future-projected climate data had 1 $\times$ 1 degree resolution.

### 2.2 Richspot schemes

The impact of climate change was assessed using three richspot schemes: (i) The "WWF Global 200" (G200) scheme of Olson and Dinerstein (2002) is the most comprehensive and was designed to represent areas prioritized for conservation on land, freshwaters and ocean; (ii)  the partly overlapping group of terrestrial "hotspots" (hereafter called Myers) proposed by Myers et al. (2000) with the modifications of Mittermeier et al. (2011) and Noss et al. (2015), which is based on species endemism and habitat loss; and (iii) the 30% highest marine biodiversity areas of Zhao et al. (2020). Results based on air temperature and precipitation were used to assess terrestrial and freshwater, and ocean temperature data were used to assess marine richspots.



### 2.3 Analysis of climate-change variables

Prior to the analyses, the climatic data layers were resampled to $0.25 \times 0.25$-degree resolution using the bilinear method, which was necessary to ensure that adequate (albeit smoothed) information is passed to small richspots. All spatial data items (climate variables and richspot schemes) were projected to the Mollweide equal-area projection, which was used throughout the

analyses. This step ensured that every pixel represents an equal area, so pixel counts translate to cumulated area and global means are not biased by the unequal spatial sampling along latitudes. For air temperature and precipitation only land-based values were included in this assessment. The coordinates of richspot centroids were tabulated to assess the latitudinal patterns of their distribution and those of their characteristic impacts.

We separated our impact variables (historic temperature difference, climate change velocity, projected warming and

precipitation) to values that fall within and outside a richspot-scheme. Within and outside richspot-scheme areas were compared with their respective mean values. We also tabulated the impacts for every individual richspot of all schemes, except that of Zhao et al. (2020) which represents a single area covering 30% of the ocean. Every richspot was characterized with one mean value of the equal-area pixels that fell within its boundaries. To express the uncertainty of within-richspot climate change due to variability among individual richspots in a scheme, we executed bootstrap simulations of richspot-means and tabulated

their mean in every simulation trial using the areas of richspots as weights. Errors are reported as the standard deviations of the bootstrap distributions, based on 10 000 trials.

For estimates of historic temperature change, we tabulated the proportion of pixels in a richspot that have been warming in the past 50 years. Richspots that had more than 95% of their pixels above $+0°C$ were considered to have been significantly affected by climate change. We also tabulated the 2.5 and 97.5% percentiles of the distributions of pixels in every richspot and

contrasted these with the global and latitudinal means of the respective variables. Richspots where the global mean was above the richspot's 97.5% percentile were considered global refugia, those with the global mean below the 2.5% percentile were considered critically warming. Refugia and critically warming richspots within latitudinal bands were tabulated the same way, and only compared to the variable's value at the latitude of the richspot's centroid (Fig. 4, Table 1).

All analyses were performed in the R programming environment (R Development Core Team, 2021). Spatial calculations were

executed using the "sp" (Pebesma and Bivand, 2005), "raster" (Hijmans, 2016), and "rgeos" (Bivand and Rundel, 2020) packages, with the utilities of the GDAL library (GDAL/OGR contributors, 2021) directly, and via its R interface "rgdal" (Bivand et al., 2017). Distributions of areas were plotted using the "beanplot" package (Kampstra, 2008).

### 3 Results

#### 3.1 Observed changes

Global warming has increased air temperature of all richspots in the Myers, as well as the terrestrial and freshwater G200 in the past 50 years (Fig. 1, Table 1. Fig. 2a). On average, warming in the Myers terrestrial ($+0.91 \pm 0.07$ °C) and G200 freshwater





(+0.89 ± 0.07 °C) richspots was less than the global average increase (+1.08°C), whereas the G200 terrestrial richspots were on par (+1.04 ± 0.1 °C). Climate change velocities were slower in all three of these richspot schemes than in the areas outside them (47%, 29% and 10% less, in Myers, G200 terrestrial and G200 freshwater richspots, respectively).

Despite that 10 of the 43 marine richspots (23%) did not witness significant overall increases of ocean temperature (Okhotsk Sea, Galápagos, Humboldt Current, Fiji Barrier Reef, Benguela Current, Agulhas Current, Rapa Nui, Patagonian Southwest Atlantic, New Zealand Marine, Antarctic Peninsula and Weddell Sea, Fig. 1a), marine richspots on average have been affected more by climate change than terrestrial or freshwater richspots. Ocean temperature in the G200 marine richspots has increased 41% more than outside (0.53 ± 0.06 vs. 0.38, with global average of 0.39 °C) and climate velocities were 69% higher than

areas outside (11.24 ± 1.86 vs. 6.64 km decade$^{-1}$, Fig. 2a). This difference is less pronounced when the RBA of Zhao et al. (2020) is considered: this area faced 4% more warming (0.41 vs. 0.39 °C) and climate velocities have been 33% larger than outside (8.86 vs. 6.65 km decade$^{-1}$).

### 3.2 Projected changes

Near-surface air temperature is projected to warm considerably faster over land than over the seas (+2.03, +2.64 and +3.93 °C

with +1.5, +2 and +3°C of warming). This means air temperatures above land warm more than over the ocean and the global average. However, when compared to other land areas (Antarctica included), freshwater richspots will be less affected by temperature changes, and will see 16, 15 and 14% less warming than areas outside at the +1.5, +2 and +3°C warming stages, respectively (Fig. 3). Terrestrial richspots defined using the Myers scheme are projected to experience similar patterns (20% less within than outside with all three stages), whereas areas using the terrestrial G200 are projected to be about as much

affected by temperature changes as areas outside them (5% less than outside).
Marine richspots of the G200 will continue to be more affected (12, 13 and 13%) than outside areas, with highest projected warming in the northern, and lowest in the southern, polar regions. The 30% RBA of Zhao et al. (2020), on the other hand, is expected to be only 1% more affected by global warming as other areas.
Global precipitation on land is expected to increase by 20, 31 and 46 mm year$^{-1}$ with +1.5, +2 and +3°C of warming. Lower-

than-outside increases are expected in precipitation in the terrestrial and freshwater richspots with each projected warming level: Myers: 128, 67 and 35% less increase; G200: 43, 12 and 5% less increase; freshwater: 59, 19, 18%, respectively.

### 3.3 Variation across richspots

Compared to the global mean temperature changes (both observed and future), most terrestrial and freshwater richspots represent climate refugia (Table 1a, Fig. 4) and only a minority of these (< 20%) are expected to warm critically (Table 1b).

In contrast to terrestrial and freshwater richspots, most marine ones of the G200 are not climate refugia, with the notable exception of the Antarctic richspot (Figure 1), and a considerable number of marine richspots are positioned in high-velocity areas. Almost half of the marine G200 are expected to face higher-than-global warming in this century.



The latitudinal distribution of richspots is similar in all three environments (see Supplementary Information). Northern high
latitude richspots will warm most, whereas the Southern Ocean and the upwelling on the Atlantic coast of southern Africa will
cool (Figs. 1, 4). Following the latitudinal patterns of warming, richspots in the northern hemisphere are disproportionally
more affected by the magnitude of temperature increase than those in the southern hemisphere (Fig. 4). Terrestrial and
freshwater richspots tend to occur in places where climate velocities are comparatively lower than those suggested by the
latitudinal average (Fig. 4b).

## 4 Discussion

Our results show that although the impacts of climate change have been lower in terrestrial and freshwater richspots, they have
been and are projected to be affected by climate change. Marine biodiversity richspots have and are projected to experience
greater effects of climate change than other areas. This discrepancy reflects both the spatial distribution of richspots and the
latitudinal patterns of climate change. The latitudinal imbalance (i.e. polar amplification) of global warming is expected to
further exacerbate the already asymmetric human impact on the marine environment and the biota (Halpern et al., 2015;
Sydeman et al., 2021).

Although overall warming is expected to affect marine richspots only slightly more based on future projections, the velocity
of climate change is extremely high in tropical richspots. Species have already responded to these changes by shifting their
latitudinal distributions polewards (Lenoir et al., 2020), which has already led to the loss of thousands of marine species from
equatorial latitudes and increases in species richness in the subtropics (Chaudhary et al., 2021).

The high climate velocities in marine richspots seem to contradict the previously suggested relationship between climate and
endemism based on long-term climate change velocity (Sandel et al., 2011). In comparison to terrestrial and freshwater areas,
the distribution of biodiversity in the ocean is more influenced by environment conditions than geographic isolation; on land
old islands and fragmented landscapes have led to higher terrestrial than marine endemism (Costello and Chaudhary, 2017;
Costello et al., 2017). Also, climate change today is happening on much shorter time scales than what may have influenced
the evolutionary origin of richspots and the distribution of endemics.

It is also possible that the definition of older richspot schemes is not representative of true biodiversity. The G200 richspots
was partly driven by political priorities ("make every nation a stakeholder", Olson and Dinerstein, 2002), and the Myers et al.
(2000) richspots were also prioritized based on threat from other human impacts in addition to their rich biodiversity. The
systematically lower difference between warming inside and outside the RBA of Zhao et al. (2020) compared to the marine
G200 might suggest that the former grasps patterns of endemism and species distributions better than the latter (Fig. 1).

The present study did not consider annual variation and additional climatic variables (Fick and Hijmans, 2017) that might
influence the distribution of species. Small-scale climate refugia might exist within the individual richspots which are not
detected due to the spatial scale of our analyses. Where there is high heterogeneity of climate change velocities (e.g., due to
topographic variation) at a spatial resolution finer that the used in our analysis, species may find thermal refugia within



terrestrial and freshwater richspots. Thus, projections as used here need to be validated by in situ monitoring of changes in species abundance.

## 5 Conclusions

Our findings support the hypothesis that most terrestrial richspots have been climate refugia in a relative sense, but they do not relax concerns regarding the effects of global warming on endemic life. While thousands of species are shifting their
geographic ranges rapidly in response to a warming climate, there is a high risk that endemic species will not be able to disperse to more suitable climates and go extinct (Manes et al., 2021). Climate mitigation is thus essential to keeping climate warming to less than 2 °C because this will reduce extinction risk in all richspots (Manes et al., 2021).

Assessment of the impact of climate change on biodiversity richspots is compounded by human-induced losses of species and habitats across all environments. As stated repeatedly in the scientific literature for decades, strict protection of biodiversity
from local human impacts within richspots is a most area-effective way to minimize species extinctions (e.g. Mittermeier et al., 2011; Darwall et al., 2018; Zhao et al., 2020). In addition, environmentally sustainable practices inside and outside richspots must facilitate species dispersal between habitats as climate change occurs.

## Code and data availability

Past climate data are openly available from the website of the MetOffice Hadley Centre
(https://www.metoffice.gov.uk/hadobs/hadisst/) and Climatic Research Unit (University of East Anglia, https://crudata.uea.ac.uk/cru/data/hrg/). Results of the CMIP6 climate data will be made publicly available from the IPCC Atlas of the AR6 report. Richspot definition schemes are available from the WWF (https://www.worldwildlife.org/publications/global-200), Zenodo (http://doi.org/10.5281/zenodo.3261807) and (https://www.sciencedirect.com/science/article/pii/S0006320719312182#ec-research-data). Coastlines were plotted using free
vector data from Natural Earth (www.naturalearthdata.com). Used data and the analytical code are archived on Zenodo along with supplementary display items and the results used to plot figures (Kocsis et al., 2021).

## Author contributions

This paper arose from discussions within IPCC WGII Cross Cutting Chapter on Biodiversity Hotspots. The study was designed by MJC and WK. ÁTK and MJC drafted the first versions of the manuscript. ÁTK carried out all the analyses and finalized
the manuscript. All authors contributed to the discussions and revisions of the paper.



**Acknowledgements**

The research was supported by the Deutsche Forschungsgemeinschaft (Ko 5382/2-1 and Ki 806/15-2) and is part of the DFG Research Unit TERSANE 2 (FOR 2332).

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





**Table and figures**

**Table 1. The percentage of richspots in each environment that have (a) the global and latitudinal mean above 97.5% of the values within them (global – latitudinal refugia) and (b) those that have the mean below the 2.5% percentile (globally – latitudinally warming). The number of richspots considered is shown in parentheses. The Lord Howe and Norfolk Island richspot of the Terrestrial G200 is not included in the assessment due its small size.**

|  | Myers (36) | G200 terrestrial (141) | G200 freshwater (53) | G200 marine (43) |
|---|---|---|---|---|
| **(a) % Refugia (global - latitudinal)** |  |  |  |  |
| Since 1971-1980 | 53 - 22 | 70 - 32 | 55 - 28 | 16 - 16 |
| Climate change velocity | 28 - 25 | 40 - 39 | 21 - 21 | 26 - 12 |
| Future +1.5°C | 69 - 42 | 73 - 46 | 66 - 40 | 5 - 26 |
| Future +2°C | 69 - 42 | 74 - 47 | 66 - 42 | 5 - 23 |
| Future +3°C | 69 - 42 | 74 - 46 | 68 - 43 | 7 - 23 |
| | | | | |
| **(b) % Critically warming (global - latitudinal)** |  |  |  |  |
| Since 1971-1980 | 8 - 17 | 9 - 20 | 13 - 19 | 28 - 9 |
| Climate change velocity | 0 - 0 | 0 - 0 | 2 - 2 | 12 - 2 |
| Future +1.5°C | 3 - 3 | 8 - 6 | 9 - 8 | 42 - 21 |
| Future +2°C | 3 - 6 | 9 - 8 | 9 - 9 | 51 - 21 |
| Future +3°C | 3 - 6 | 9 - 7 | 8 - 11 | 49 - 19 |




**Figure 1. Recorded global warming in the terrestrial, freshwater and marine environments in the past 50 years. (a) The mean absolute changes (°C) in richspots between the average annual means between the 1971–1980 and the 2011–2020 interval, (b) spatial distribution of climate change velocities (km decade⁻¹). Terrestrial and freshwater richspots are assessed with near-surface air temperatures, ocean surface temperatures were used with marine richspots. Note the high spatial variability of climate change velocities.**



Figure 2. Recorded patterns of global warming in richspots of the terrestrial, freshwater and marine realm. (a) The difference between 1971–1980 and 2011–2020, and (b) velocities of climate change in the same interval. Beanplots show the distribution of area (density of equal area cells) in the richspot schemes.





**Figure 3. Future-projected temperature change and precipitation using the CMIP6-based scenarios at stages of +1.5, +2 and +3°C global warming. Beanplots show the distribution of area (density of equal area cells) in the richspot scheme. Solid black lines indicate the land- or ocean-based global means, dashed lines indicate mean value outside and inside the hotpots.**

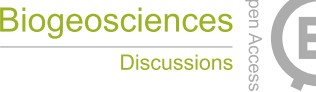

**Figure 4. Latitudinal patterns of global warming in individual richspots. a. Recorded absolute changes between the average annual means in the 1971–1980 to the 2011–2020 interval, b. climate change velocities in the same interval, c. projected warming compared to pre-industrial conditions when warming reaches the +1.5, +2 and +3°C levels (averaged across multiple scenarios). Dashed lines indicate global means (only land or ocean, respectively), solid curves indicate the latitudinal means. Vertical bars denote the interval between the 2.5 and 97.5% percentile of values within one richspot. Triangles indicate richspots that are critically warming compared to the global mean, diamonds indicate global refugia. See Table 1 for the tabulation of refugia and critically warming richspots.**