# Peer review of "Not all biodiversity richspots are climate refugia"

_Biogeosciences, 2021_

## Author Response (AR1)

We have answered the questions of the Referees and addressed their issues in the manuscript. Please see the text below for a point-by-point summary.

**Response to Referee 1**

Referee: This is a well-written manuscript that describes an analysis of the correlation between biodiversity richspots and climate change refugia. The results are not surprising but still good to have as evidenced.

I only have a few comments, mostly have the presentation of the results.

**Answer: We are thankful for the kind words and the constructive feedback.**

Referee: The opening of section 3.3 seems like an important conclusion that is unnecessarily downplayed in the abstract (although the marine part is appropriately highlighted there). Contrast the wording there to this in the abstract, which almost seems like spin that does not appropriately represent the results, "The results suggest that although terrestrial and freshwater richspots have been and will be somewhat less affected than other areas, they are not excluded from the impacts of global warming. Their characteristic biota is expected to witness similar forcing as other areas, including range shifts and elevated risk of extinction." It seems like this is worded to get attention to a misunderstanding that I don't/didn't think exists - that people think that climate refugia, and thus biodiversity hot/richspots, and thus perhaps biodiversity?, will be completely unaffected by climate change.

The message that refugia will warm is not novel; indeed this has been repeatedly stated in the literature (eg see Morelli et al. 2020 or any of the papers in the Special Issue on Climate-Change Refugia in Frontiers in Ecology and the Environment). It is probably worthwhile to show that globally though.

**Answer: Our phrasing in the abstract reflects on the initial null hypothesis that richspots are expected to represent refugia, which we tested. Several of the cited studies stated this was the case but they only studied a limited selection of such richspots. Given that we provide some evidence that this might be true to some extend in some locations at least (the majority of richspots in the terrestrial and freshwater realms are in fact expected to face less pronounced changes) we found it noteworthy to be mentioned in the abstract.**

**However, we agree, and will change the wording of the abstract to better reflect the current consensus regarding the effects of climate change in biodiversity richspots. This will also be emphasized in the main text by including some of the papers that the Reviewer mentioned (e.g. Morelli et al.).**

**Changes: The abstract was rephrased to better reflect the current consensus regarding the effects of climate change in biodiversity richspots. We included new papers in the manuscript from the issue that the Referee suggested (Morelli et al. 2020, and Michalak et al. 2020; e.g. lines 84 and 215).**

Referee: The difference between marine and terrestrial richspots is likely (also) related to what drives diversity in marine versus terrestrial systems, where connectivity is so different and geography often differing drastically between life stages. There is a short mention on line 192 but I suggest saying more. The connection to tropical marine diversity is mentioned but the distinction between processes there (187-190) and in the terrestrial tropics is not clear.

**Answer: Including more information about the differences among realms is a good idea (for example, relative endemicities across the three environments, which reflect their connectivity, might lead to different reactions of richspots to climate change). We will expand this section somewhat during the revisions**

**Changes: We added additional points about the likely differences in the drivers of biodiversity in the realms, mentioning the filling of fundamental niches and habitat connectivity. The endemicity values of the different realms are now mentioned in this paragraph as well. See lines 195-204.**

Referee: A few minor suggestions:

107: Move "(hereafter called Myers)" to after (2015)

120: into not to

155: seems a bit repetitive

156-159: This is a bit hard to follow. Suggest listing the Terrestrial richspots…part before talking about freshwater. Also not clear what "less within than outside with all three stages" means

161: suggest rewording as "with highest and lowest projected warming in the northern and southern polar regions, respectively.

**Answer: We are grateful for the suggested changes in the wording and will incorporate these in the revised version of the manuscript.**

**Changes: The suggested changes were incorporated in the text.**

**Response to Referee 2**

Referee: In their manuscript entitled "Not all biodiversity richspots are climate refugia", Kocsis et al. explore the vulnerability of marine, terrestrial and freshwater biodiversity richspots (regions that currently have exceptionally high endemism and/or number of species) to anthropogenic climate change. They specifically test whether these regions are likely to provide refugia under future climate change. The results bring valuable evidence for the need to protect a representative network of biodiversity-rich areas that also considers species' vulnerability to climate change.

The paper is well written, and I enjoyed reading it. The analyses are straightforward and use sound methodology. In fact, I have virtually no concern with seeing the paper published in its current form. I will only make a few suggestions to improve the manuscript, which the authors can choose to consider.

**Answer: We are thankful for the kind feedback and are happy to implement the suggestions.**

Referee: It is somewhat confusing that "climate velocity" is interchangeably used in the manuscript to refer to long-term climate change since the last glacial maximum and recent/future anthropogenic climate change velocity over a few decades. I understand that both concepts refer to similar mechanisms and are called like this in the literature (e.g. Sandel et al. 2011, Loerie et al. 2009), but as mentioned in L.194, the consequences are very different, owing to the different time scale considered (evolutionary origin of richspots and driver of endemism in the first case, and drivers or range shifts and vulnerability to anthropogenic climate change in the second case). Maybe it could be clarified early in the text that this analysis focus on the latter, also avoiding the use of the term interchangeably?

**Answer: Indeed, the term "climate velocity" has been used by us and by the community in a way that can refer to both long-term and short-term changes. We do not consider this to be bad practice, as these references reflect the same physical dimension, albeit on different scales. However, it would indeed be preferable make these references more explicit. We will try to clarify whether climate velocity in a context refers to long-term or short-term changes.**

**Changes: The qualifier and 'short-term' was added when climate velocities were mentioned in a dubious contexts (e.g. line 57). We also added a sentence to bring the attention of the reader to the issue of time scales when calculating climate velocities (line 55).**

Referee: At the end of the introduction, the authors mention that these "findings are of importance for policies to address biodiversity loss and climate change". However, the discussion does not further mention the relevance for policy and biodiversity management, except calling for climate change mitigation and mitigation of other threats in the conclusion. Maybe the authors could further develop the implications of their findings for the identification of conservation priority areas and management of biodiversity richspots. How are these currently used in decision making and how are these findings relevant to better prioritize biodiversity conservation in a changing world?

**Answer: We will elaborate the implications of our results on conservation management. For example, we can explain how the absence of the analysis presented in this paper, leaves the issue of whether all or some biodiversity richspots will be to any extent natural refugia from climate change unanswered. As these areas represent most of the world's biodiversity it may thus be speculated that biodiversity may not be as badly affected by climate change as studies reporting results on taxa in regions subject to rapid warming.**

**Changes: We added another paragraph to the discussion to help better place the implications of our study into context. See lines 216-222.**

Referee: Huntley & al (2021) recently explored (terrestrial) biome consistency in the past and under projected anthropogenic climate change. This study was not published at the time of submission, but it seems that it could now be incorporated to discuss how these two approaches support each other in their findings and conclusions.

Huntley, B. et al. 2021. Projected climatic changes lead to biome changes in areas of previously constant biome. - J Biogeogr: DOI: 10.1111/jbi.14213

**Answer: We are thankful for the suggestion and will incorporate the paper by Huntley et al. in our study.**

**Changes: The suggested paper was incorporated into the study (e.g. line 64).**

Referee: Minor comments:

20-21: This sentence could be simplified to "It has been suggested that some geographic areas (…) because they have had (…)"

77-78: There is no mention of invasive species anywhere else in the paper, which makes this sentence quite out of topic. Could be deleted?

56: Missing dot in "distance.time-1"

**Answer: The suggested corrections will be incorporated into the manuscript. Again, we are very grateful for the feedback.**

**Changes: The suggestions were incorporated in the text.**